# Water-Based Generators with Cellulose Acetate: Uncovering the Mechanisms of Power Generation

**DOI:** 10.3390/polym16030433

**Published:** 2024-02-04

**Authors:** Seung-Hwan Lee, Hyun-Woo Lee, So Hyun Baek, Jeungjai Yun, Yongbum Kwon, Yoseb Song, Bum Sung Kim, Yong-Ho Choa, Da-Woon Jeong

**Affiliations:** 1Korea National Institute of Rare Metals, Korea Institute of Industrial Technology, Incheon 21655, Republic of Korea; leesh93@kitech.re.kr (S.-H.L.); totptkd12@kitech.re.kr (H.-W.L.); qorthgus9@kitech.re.kr (S.H.B.); yjj0011@kitech.re.kr (J.Y.); kyb916@kitech.re.kr (Y.K.); songys88@kitech.re.kr (Y.S.); bskim15@kitech.re.kr (B.S.K.); 2Department of Materials Science and Chemical Engineering, Hanyang University, Ansan 15588, Republic of Korea; choa15@hanyang.ac.kr

**Keywords:** cellulose acetate, eco-friendly generator, water, generation mechanisms

## Abstract

Power generation technologies based on water movement and evaporation use water, which covers more than 70% of the Earth’s surface and can also generate power from moisture in the air. Studies are conducted to diversify materials to increase power generation performance and validate energy generation mechanisms. In this study, a water-based generator was fabricated by coating cellulose acetate with carbon black. To optimize the generator, Fourier-transform infrared spectroscopy, specific surface area, zeta potential, particle size, and electrical performance analyses were conducted. The developed generator is a cylindrical generator with a diameter of 7.5 mm and length of 20 mm, which can generate a voltage of 0.15 V and current of 82 μA. Additionally, we analyzed the power generation performance using three factors (physical properties, cation effect, and evaporation environment) and proposed an energy generation mechanism. Furthermore, we developed an eco-friendly and low-cost generator using natural fibers with a simple manufacturing process. The proposed generator can contribute to the identification of energy generation mechanisms and is expected to be used as an alternative energy source in the future.

## 1. Introduction

As the world population is rapidly increasing and living standards become higher, energy demand continues to increase. This has resulted in global research efforts aimed at bolstering energy supply [1]. However, the traditional approach of relying on fossil fuels for energy generation is characterized by the emissions of various pollutants, including particulate matter and ozone. These pollutants have resulted in severe health problems such as heart disease, respiratory illnesses, and vision impairment. Additionally, the finite nature of fossil fuel reserves and the diminishing availability of economically viable and high-quality oil sources have contributed to a significant increase in production costs [2,3].

Conventional methods for generating eco-friendly renewable energy, such as solar power, are constrained by limited generation time (an average daily sunlight duration of approximately 3.6 h) and the requirement for a broad area because of the inability to stack p–n junction semiconductors that rely directly on the photovoltaic effect for increased capacity [4]. Similarly, wind power generation is dependent on strong wind conditions and faces geographical limitations as it is only feasible in high-altitude or offshore areas. Furthermore, the generation process emits low-frequency noise, which can affect human physiological functions such as the circulatory and respiratory systems. Moreover, the power output is proportional to the square of the blade size, necessitating large-scale installations [5].

Recent trends in energy-harvesting studies have focused on technologies that harvest energy from the surrounding environment with fewer spatial, temporal, and size constraints. Notable examples include piezoelectric/friction electricity [6,7], photovoltaic devices [8], and evaporation-driven power generation. Evaporation-driven generation technology, in particular, leverages water, which covers more than 70% of the Earth’s surface and is universally available, offering relatively fewer time and space restrictions than other generation methods [9,10,11,12,13,14,15,16]. In addition, other research organizations are also investigating various materials for generator materials, which is expected to lower the cost of generator production in the future [17,18,19,20,21,22,23,24,25,26].

Hydroelectric power, a method employed for energy generation for over two millennia, has recently unveiled a novel avenue for effective energy production. This innovation centers on the interaction between water and conductive particles within nanochannels powered by pressure-gradient-driven processes [27,28]. The foundation of this electricity generation is electrokinetics effects. When a charged solid surface interfaces with water, an electric double layer (EDL) is formed at the boundary [29,30]. In a confined nanoscale space, overlapping EDLs cause charged ions to occupy the majority of the space. When a pressure gradient induces fluid flow through this space, a streaming current is generated [9].

Notably, the key to energy production lies in the influence of water evaporation. The evaporation of water directly transforms heat from the surrounding environment into electricity. This concept has been substantiated by density functional theory simulations, which demonstrate electricity generation through water evaporation at the interface between water and carbon materials [31]. Numerous studies have corroborated the potential for electricity generation via diverse interactions between water and conductive nanoparticles, encompassing both streaming current and water evaporation phenomena. These findings have been translated into English, encompassing a wide array of engineering articles. Different research groups have proposed diverse mechanisms for energy-harvesting generators using water, each attributing energy generation to various contributing factors. Existing studies have revealed multiple primary principles underlying energy generation. Capillary action and surface interaction: Some studies have suggested that energy is generated through the interaction between a solid surface and flowing fluid, particularly through capillary action within micro- or nano-pores [25,32]. Evaporation-induced polarization: Another perspective focuses on the evaporation of a solution adhering to the surface of a conductive material. This natural phenomenon leads to polarization at the interface, creating a potential difference within the material and facilitating the flow of electrons [33,34,35]. Pseudo-streaming current: Additionally, research findings have indicated that energy can be harnessed through the pseudo-streaming current that arises from proton movement in conjunction with capillary water flow [28,36].

However, several critical phenomena, such as ion adsorption on solid surfaces, power generated relative to ion concentration, and the relationship between penetration direction and electricity generation, have remained unexplained. In 2023, Ko et al. conducted a groundbreaking study, wherein simulation results were validated through practical experiments and multiphysics models, shedding light on the precise mechanism through which energy is generated when porous carbon interacts with water or aqueous solutions [37]. This study elucidates the operating principles of such energy-harvesting systems. Currently, studies on water-based energy generation have progressed, primarily directed toward optimizing conductor materials and generator configurations to increase overall efficiency.

In this study, we developed an environmentally friendly energy-harvesting device that uses water and investigated its energy generation mechanism by categorizing it into three distinct types: physical properties, cation effect, and evaporation environment. Cellulose acetate, commonly found in cigarette filters, was used to fabricate the generator, and the conductive material was coated with carbon black (CB). Cellulose acetate is cost-effective and simple to manufacture. Moreover, its hydrophilic, natural, and eco-friendly properties render it a suitable candidate for eco-friendly energy-harvesting materials. CB was selected as a conductive material owing to its high electrical conductivity and large specific surface area in a hollow shape, enhancing its energy-harvesting performance, particularly in facilitating evaporation. To validate the energy generation mechanism, we varied the generator’s specifications such as size (diameter, length, and connections) to investigate its relationship with energy generated by streaming current. Additionally, we varied the type and quantity of initially injected ions to assess the influence of cation effects on performance. Further, we investigated the generator’s performance under different environmental conditions, particularly humidity levels. Through these experiments, we presented research findings that validate the energy generation mechanism using water.

## 2. Materials and Methods

### 2.1. Fabrication of a Cellulose Acetate Column Generator (CACG)

To prepare the conductive coating solution, 0.625 wt.% carbon black powder (Ketjen Black EC 600 JD, Lion Corporation, Tokyo, Japan) and 1.5 wt.% Cetyl trimethyl ammonium bromide (CTAB) surfactant were combined in DI water and put in an ultrasonic for 2 h. In nanoparticle dispersions, surfactants used as dispersants are adsorbed on the surface of the nanoparticles to reduce surface tension. The charge on the particle surface is adjusted to increase the zeta potential, which increases electrostatic repulsion and prevents agglomeration [38]. The charge on these solid surfaces is different for each surfactant, and previous studies have shown that carbon black dispersion depends on the type of surfactant [35]. CACGs are generators in which a CAC is coated with a conductive solution prepared. To analyze the energy generation mechanism, a CACG was manufactured with two distinct diameters (5 mm and 7.5 mm) and five different lengths (10, 20, 30, 40, and 50 mm), resulting in the creation of 10 types of generator designs (TAEYOUNG INDUSTRY CORP., Seoul, Republic of Korea). To achieve uniform carbon coating across the entire porous CACG surfaces via capillary action, both ends of the CACG were dipped in CB solution. The CAC was dipped vertically into the carbon black solution, and then it was turned over and dipped once more. This method ensured consistent coating on both the interior and exterior surfaces of the generator. Finally, the CB-coated CACG specimens underwent a complete drying process in a convection oven set at 80 °C for 5 h, thus finalizing the production of the CACGs, as shown in Figure 1.

### 2.2. Characteristic and Electrical Measurement of the CACG

The morphology of the CACG was analyzed using field-emission scanning electron microscopy (FE-SEM; JSM-7100F, JEOL Ltd., Akishima, Japan). Specific surface areas were determined using Brunauer–Emmett–Teller (BET) measurements (3Flex, Micromeritics Instrument Co., Norcross, GA, USA). Surface functional groups were identified using Fourier-transform infrared (FT-IR) spectroscopy (VERTEX 80V; Bruker, Billerica, MA, USA). The zeta potentials of both the solutions and solid surfaces were measured using an electrophoretic light scattering zetasizer (ELSZ-1000ZS, Otsuka Electrics and ELSZ-2000ZS, Otsuka Electrics, respectively, Osaka, Japan). The electrical resistance of the CACG was measured using a multimeter (Fluke 15b+, Fluke Corporation, Everett, DC, USA). Arduino Uno hardware (SZH-EK002, Arduino LLC, Lombardia, Italy) and a precise thermo-hydro sensor probe (FS200-SHT10, Sensirion AG, Stäfa, Switzerland) were used to monitor and record temperature and humidity data. The desired temperature and humidity conditions within the acrylic box were controlled by deploying a humidifier and a dehumidifier. All experiments were conducted under controlled conditions with room temperature and humidity level maintained at approximately 25 °C and 30%, respectively. Following the addition of 100 μL of a 2 M NaCl aqueous solution to one electrode to initiate power generation, the open-circuit voltage (*V*_oc_) and short-circuit current (*I*_sc_) were measured using a source measurement unit (Keithley 2400, Cleveland, OH, USA). Control of the measurement apparatus was facilitated using PC software (I.V Solution, Seoul, Republic of Korea), as shown in Figure 2.

## 3. Results and Discussion

### 3.1. Characterization of the Conductive Materials, CAC and CACG

Analyses were performed to characterize the chemical and physical properties of the conductive CAC and CACG solutions used to fabricate the generators. Figure 3a shows the chemical structure of CA, which contains hydroxyl groups (O–H), rendering it suitable for absorbing water [39,40]. Figure 3c shows the FT-IR analysis results of the CAC, CB, and CB-coated CAC (CACG). Cylindrical CAC samples with a diameter of 7.5 mm and a length of 20 mm were cut for FT-IR analysis. CACG samples were prepared from a conductive solution of CB (0.625 wt.%), CTAB (1.5 wt.%), and DI water. The representative chemical structure of the CAC comprises O–H (3499 cm^−1^), C–H (2923 cm^−1^), C=O (1759 cm^−1^), and CH_3_ (1390 cm^−1^). CB is a suitable conductive material owing to its O–H (3499 cm^−1^) groups, high conductivity, and specific surface area [40,41,42]. In the CACG FT-IR results, the transmittance of C–H and C=O decreased, and the presence of O–H and COO– showed that the CAC was well coated with carbon black. The coating of the carbon solution to the core can be seen in the EDS analysis in Figure 4b. CB tends to self-aggregate; therefore, its dispersion should be increased before coating. The surfactant added to stir the CB solution was CTAB, a cationic surfactant, which is suitable because it has a pole opposite to that of CB, which has a negative zeta potential [35,43,44]. The zeta potential is considered evenly distributed over ±30 mV, and the solution prepared in this study comprised CB (0.625 wt.%), CTAB (1.5 wt.%), and DI water, confirmed by the zeta potential analysis value (30.25 mV) shown in Figure 3e. In addition, as shown in Figure 3f, particle size analysis of the CB used in the coating solution showed that a nano powder of approximately 340 nm was selected as a suitable material to increase the specific surface area for the production of the CACG. As a result of measuring the particle size using the distribution intensity of the coating solution we prepared, the average diameter was 341.5 nm and the diameter (50%) was 349.4 nm. In addition, it can be seen from the original data results that it is not aggregated to a specific size and is evenly distributed at 341 nm (Appendix A). Notably, an increase in the specific surface area of the generator improves energy-harvesting performance. Specific surface area (BET) analysis of CB was 1192 m^2^/g, confirming that it has favorable properties for evaporation [45]. The specific surface area analysis was performed after a 20 h pretreatment in argon gas at 80 °C. The generator lengths were both 20 mm, and a CACG_5 with a diameter of 5 mm and a CACG_7.5 with a diameter of 7.5 mm were analyzed. After carbon coating, the conductivity and specific surface area increased from 0.725 m^2^/g for the CAC to 4.701 m^2^/g for the CACG. In addition, when the specific surface area was determined by changing only the diameter of the CAC, the specific surface areas of CACG_7.5 and CACG_5 were 4.701 m^2^/g, and 2.172 m^2^/g, respectively, showing a difference of approximately two. The following section compares the generator’s performance based on the difference in specific surface area. SEM–energy dispersive spectrometry (EDS) analysis was performed for the internal and external characterization and elemental analysis of the CACG, as shown in Figure 4. The CACG is a compact cylindrical generator with a diameter of 7.5 mm and length of 20 mm, and the diameter of the CA fibers inside is approximately 20 µm. The fibers contained pores, which enable water to capillary flow inward and evaporate outward. In addition, the coating method proposed in this study was verified by SEM analysis, in which CB was coated at the center of the CACG, as shown in Figure 4a. An EDS mapping analysis was performed to determine whether the carbon solution was evenly coated inside the CACG, as shown in Figure 4b. As CB and CA both contain C and O, they cannot easily be distinguished. Therefore, the degree of coating was determined using N and Br, which are the elements of the CTAB surfactant contained in the CB solution (Figure 3a). Additionally, in the SEM image in Figure 4c, it was confirmed that the end part was coated with 200 μm and the center part was also coated with carbon powder. In fact, as a result of measuring the resistance of 10 CACG (D: 7.5 mm, L: 20 mm) generators manufactured with one coating, most were manufactured around 2 kΩ, verifying the reproducibility of the generator manufacturing. Based on the characteristics of the CACG, we analyzed the energy-harvesting mechanism and cross-validated the power generation performance under different conditions (length, injection solution, and humidity).

### 3.2. Mechanism of the CACG for Electricity Generation

The energy-harvesting mechanism using water proposed in this study has three components. The physical properties of the CACG include the cationic nature of the injected solution and finally the environment (relative humidity) in which it was developed.
*I*_Total energy_ = *I*_Physical properties_ (*I*_P_) + *I*_Cation effect_ (*I*_C_) + *I*_Evaporation environment_ (*I*_E_)(1)

Figure 5 shows a schematic of the development mechanism of the CACG, which was fabricated by mimicking the process of transpiration, which is the uptake of water from the roots of a plant, diffusion through the stem, and evaporation through the leaves [46]. The process of energy harvesting was initiated by adding 100 µL of 2 M NaCl to one end of a completely dry CACG. When water is absorbed in one end, capillary action naturally occurs between wet and dry parts and a pressure difference exists, inducing *I*_P_. In addition, depending on the amount and type of the injected solution (2 M NaCl), *I*_C_ is induced, energy is generated by evaporation [22,47], and *I*_E_ is induced by the external evaporative environment, explaining the complex harvesting mechanism. The experimental conditions of the electrical performance were as follows: the CAC was 20 mm in length and 7.5 mm in diameter, 100 µL of 2 M NaCl was used, and there was 30% relative humidity. All experiments were conducted five times, and the standard deviation was expressed using three results, excluding the highest and lowest data.

#### 3.2.1. Mechanism of the CACG Energy by Physical Properties

The rationale for using the CACG to convert evaporative energy into electricity is based on the combination of theoretical analysis and simple estimation of the flow potential/current, capillary pressure, and hydrodynamic flow resistance equations. Consider a capillary of length *L*, pore cross-sectional area *A*, and internal surface zeta potential ξ. When a solution is pumped through the CACG by an external pressure difference Δ*P*, the streaming potential can be expressed as follows [48,49]:(2)$$I_{s}=\frac{A\varepsilon_{o}\varepsilon_{r}}{\eta L}\Delta P\xi ,$$
where $\varepsilon_{o}$, $\varepsilon_{r}$, *σ*, ξ, and η are the dielectric constant, conductivity of the fluid-saturated porous medium, surface zeta potential, and liquid viscosity, respectively. The capillary pressure expressed by the Young–Laplace equation mainly corresponds to the Δ*P* between the top and bottom surfaces of the CACG. Therefore, the capillary pressure applied by the CACG is Δ*P* = 4*γ* cos*θ*/*d*, where *θ* is the water contact angle of the CACG, *γ* is the surface tension of the water, and *d* is the diameter of the individual CACG pores. By substituting Δ*P*, we derive the following equation for the flow potential.
(3)$$V_{s}=\frac{4\varepsilon_{o}\varepsilon_{r}}{\eta \sigma d}\xi cos\theta$$

Using the aforementioned theoretical formulas, a performance analysis was conducted by adjusting the diameter and length of the CACG to optimize the physical properties of the generator. All performance data are averaged for 200–500 s of performance following the injection of 100 µL of 2 M NaCl solution (Appendix A). As shown in Figure 6a, the length of the CACG is fixed at 20 mm, the diameter is varied between 5.0 and 7.5 mm, and the change in power generation performance due to diameter variation is analyzed. For CACG_5.0 (diameter of 5.0 mm), *V*_oc_ and *I*_sc_ are 0.040 V and 42.5 μA, respectively, and for CACG_7.5, *V*_oc_ and *I*_sc_ are more than twice as high at 0.147 V and 81.9 μA. This is because of the increase in area (*A*) with increasing diameter, increasing the value of *I*_streaming_ (Equation (1)), and the increase in specific surface area (Figure 3d), improving the power generation by evaporation. In addition, as the diameter increases, the number of channels through which internal electrons move increases, reducing resistance and thus self-loss. Figure 6b compares power generation performance as a function of length for a generator with a diameter of 7.5 mm and varying lengths of 10, 20, 30, 40, and 50 mm. Notably, the resistance of a typical resistor increases with increasing length, and this increase indirectly verifies that the coating is uniform [50]. The low power generation performance at the length of 10 mm is because the Δ*P* value is extremely low as the length is extremely short when the NaCl solution is injected. Therefore, the *I*_s_ value is extremely low, and after 20 mm, Δ*P* is constant. However, the power generation performance decreases as the length increases. Therefore, the optimal CACG length for a fixed injection of 100 µL of 2 M NaCl is 20 mm, implying that a short-length generator can produce an energy harvester.

#### 3.2.2. Mechanism of the CACG Energy by Cation Effect

The second mechanism of energy harvesting is a performance analysis based on the type and amount of injected solution. Water-based generators require an initial solution injection. In this case, comparable characteristics are the effects of the ions applied, either without ions, such as DI water, or with ions, such as NaCl, which was the solution mainly used in this study. In addition, the principle of power generation can be analyzed by varying the amount of injected solution, such as a 2 M NaCl solution. The difference in performance between DI water and NaCl solution, as shown in Figure 7a, is significant and is attributed to the energy generation mechanism involving the EDL of the cations in the injected electrolyte. In previous studies, conductive carbon materials with large surface areas were reported to spontaneously generate EDLs to reduce surface energy when immersed in liquids with Equations (2) and (3). Furthermore, non-polar liquids and polar non-protonated liquids do not generate streaming potential/current-induced electricity regardless of their high dielectric constant. However, polar protonated liquids have been found to generate electricity [23,25,26,46,51,52]. Figure 7b shows a generator with dimensions of D: 7.5 mm and L: 20 mm, and the generator performance was compared as a function of initially injecting 10, 50, 100, 200, and 300 µL of 2 M NaCl. The results of the initial injection solution volumes of 10, 50, and 100 µL show that both voltage and current increase as the volume of ions injected increases, validating the improvement in generator performance with the volume of ions. However, at 200 and 300 µL, the performance decreased because the Δ*P* value decreased due to the large amount of solution relative to the length of the generator, resulting in less energy generation by streaming current. These results suggest that generator optimization requires a combination of the physical properties of the generator and injection solution volume.

#### 3.2.3. Mechanism of the CACG Energy by Relative Humidity and Multiple Connections

Finally, to compare the energy-harvesting performance by evaporation, we controlled the external humidity. All previous performance measurements were conducted in a 30% humidity atmosphere. Figure 8a shows an increase in external humidity to 50% and 80%. Consequently, the solution injected into the generator could not easily evaporate into the atmosphere. Because the CACG characteristics and the type and amount of the injection solution were constant, *I*_P_ and *I*_C_ involved in energy generation were the same, and the degree of energy generation by *I*_E_ was comparatively analyzed. In the 30% humidity condition optimized in this study, the power generation performance was 0.15 V and 82 μA for *V*_oc_ and *I*_sc_, respectively, but the performance at 80% was reduced to 0.085 V and 52 μA, respectively, which is approximately one third of the performance. Therefore, to optimize the generator, the physical properties, injection solution conditions, and evaporation environment are essential for improving the energy generation performance. As shown in Figure 8b, similar to the case of commercial batteries, connecting multiple devices in series or parallel can enhance the voltage and current outputs of the device. Therefore, for applications in devices such as light-emitting diodes, the voltage current can be easily increased using series–parallel connections.

## 4. Conclusions

In this study, we developed an eco-friendly water-based energy-harvesting device employing cellulose acetate, a readily accessible and cost-effective material found in cigarette filters, as a generator. CB was used as the conductive material owing to its excellent electrical conductivity and large surface area, which are conducive to enhanced energy harvesting, particularly through evaporation. The optimal condition of the developed CACG is a cylindrical generator with a diameter of 7.5 mm and length of 20 mm, with an injection solution of 100 µL of 2 M NaCl and relative humidity of 30%, which can generate a voltage and current of 0.15 V and 82 μA, respectively. The energy generation mechanism was systematically explored by categorizing it into physical properties, cation effects, and environmental evaporation variables. By varying the specifications of the generator and ion injections, we elucidated their impact on energy generation, particularly the streaming current, and probed their performance under varying humidity conditions. These comprehensive experiments validate the potential of water as a reliable source of sustainable energy and offer a promising avenue for eco-friendly energy-harvesting applications.

## Figures and Tables

**Figure 1 polymers-16-00433-f001:**
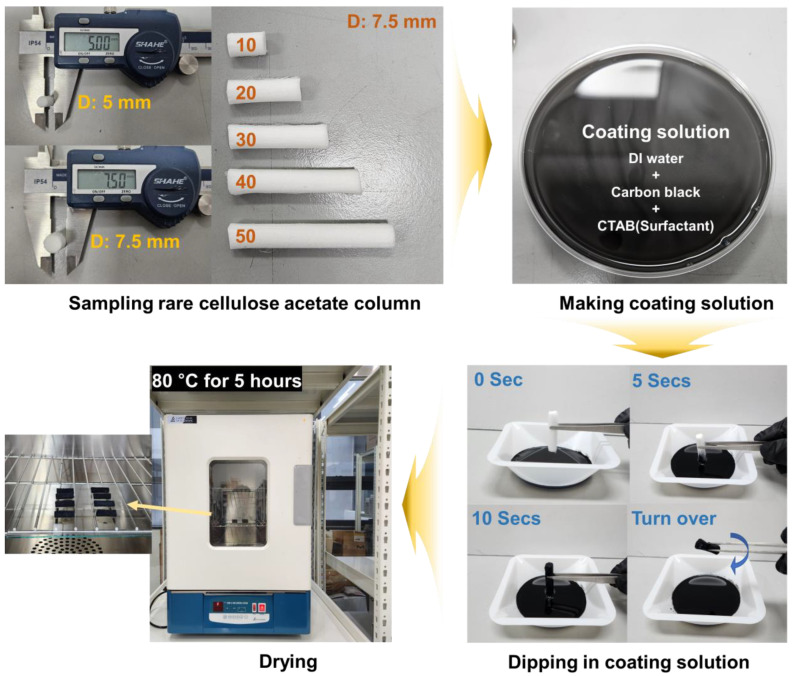
Process sequence real images of rare cellulose acetate column (CAC) sampling: coating solution preparation, coating method, and drying process.

**Figure 2 polymers-16-00433-f002:**
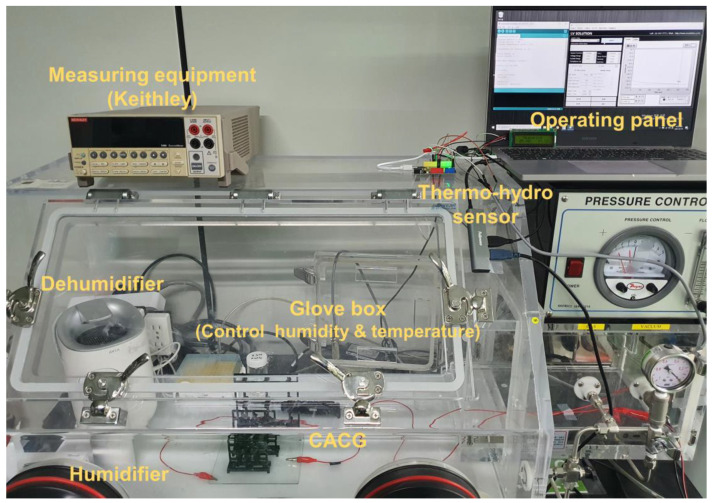
Experimental setup for measuring cellulose acetate column generator (CACG) performance in a temperature- and humidity-controlled acrylic box.

**Figure 3 polymers-16-00433-f003:**
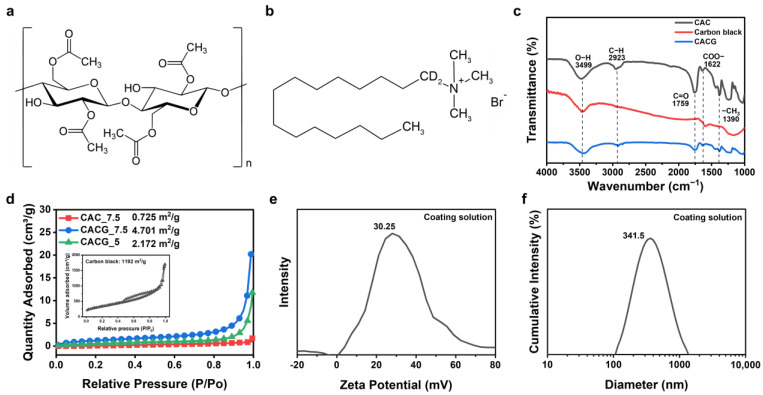
(**a**) Chemical structure of cellulose acetate and (**b**) cetyltrimethylammonium bromide. (**c**) Fourier-transform infrared spectra of CAC, carbon black (BC), and CACG. (**d**) N_2_ adsorption isotherms of the CAC, CACG, and CB. (**e**) Zeta potential results of coating solution. (**f**) Intensity distribution results of coating solution.

**Figure 4 polymers-16-00433-f004:**
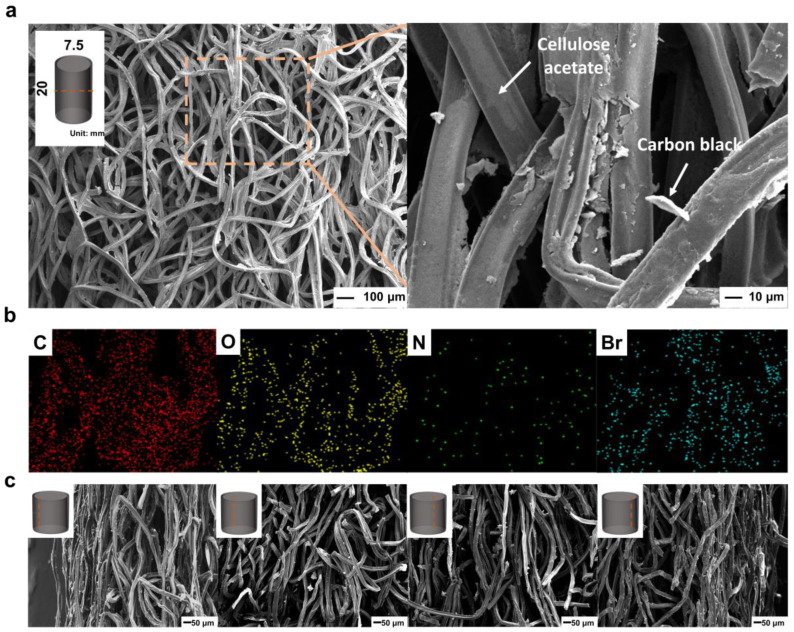
(**a**) SEM image of cellulose acetate column generator. (**b**) EDS mapping for each element (C, O, Br, and N) of the cellulose acetate column generator. (**c**) SEM images of the cellulose acetate column generator by location (left, middle 1, 2, right).

**Figure 5 polymers-16-00433-f005:**
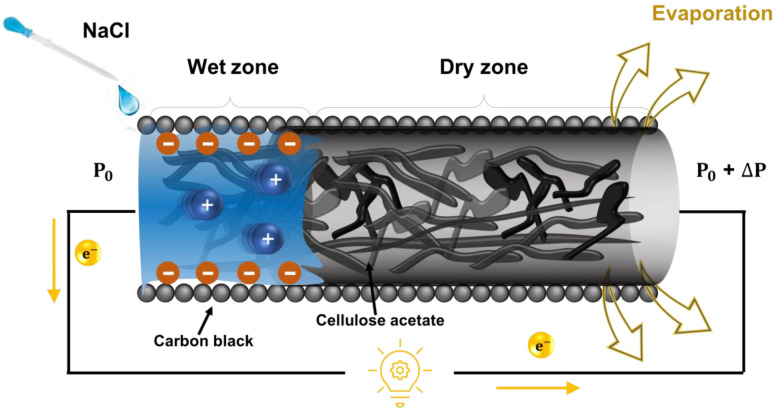
Schematic of the energy generation mechanism of water-powered generator. (Yellow arrow, Direction of electron movement).

**Figure 6 polymers-16-00433-f006:**
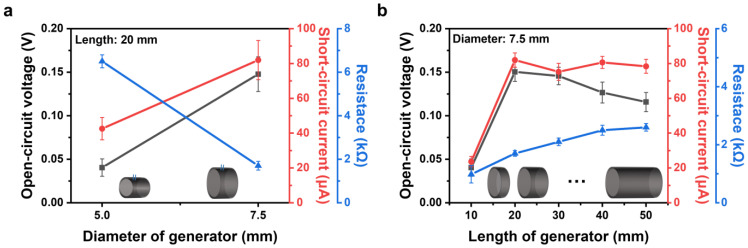
CACG performances (*V*_oc_, *I*_sc_) and resistance based on physical properties. (**a**) Same length: 20 mm; diameter: 5 and 7.5 mm. (**b**) Same diameter: 7.5 mm; length: 10, 20, 30, 40, and 50 mm.

**Figure 7 polymers-16-00433-f007:**
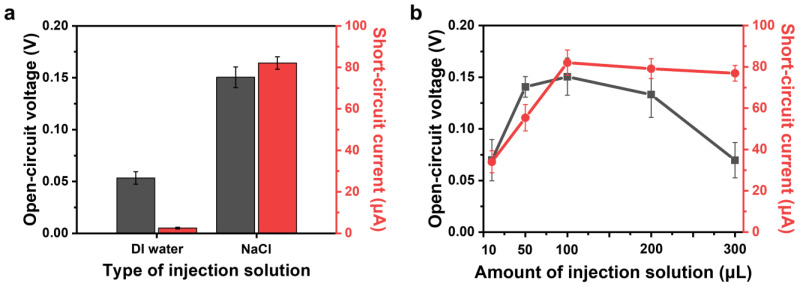
CACG performance (*V*_oc_, *I*_sc_) based on cation effect (length: 20 mm; diameter: 7.5 mm). (**a**) Injection solutions consist of DI water and NaCl; (**b**) performance depending on the amount of NaCl in the injection solution.

**Figure 8 polymers-16-00433-f008:**
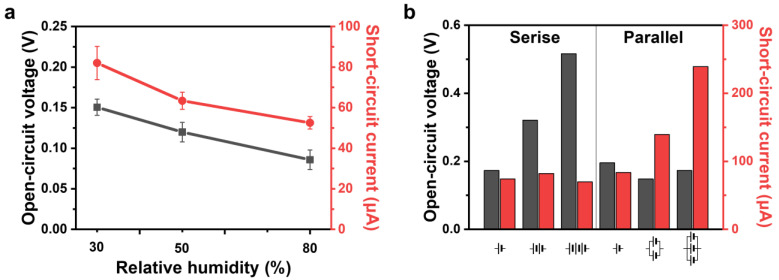
CACG performance (*V*_oc_, *I*_sc_) based on cation effect (length: 20 mm, diameter: 7.5 mm). (**a**) Measurement of performance changes due to relative humidity (30%, 50%, and 80%). (**b**) Series and parallel connections of the CACG.

## Data Availability

Data are available in the main article and Appendix A.

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
