# Peer review of "Water-Based Generators with Cellulose Acetate: Uncovering the Mechanisms of Power Generation"

_polymers, 2024, doi:10.3390/polym16030433_

Round 1

Reviewer 1 Report

Comments and Suggestions for Authors

Da-Woon Jeong et al., in their manuscript entitled "Water-Based Generators with Cellulose Acetate: Uncovering the Mechanisms of Power Generation," prepared a water-based generator by coating cellulose acetate with carbon black. The authors analyzed the power generation performance and proposed the energy generation mechanism. However, some of the limitations listed below were observed:

  1. The manuscript contains many typos and spelling mistakes. They should be corrected.
  2. The introduction is general. The authors should improve it by adding more information and appreciated references about similar works to their proposed generator, with cons and pros.
  3. The authors should explain the amount of Ketjen Black powder (0.5 g), hexadecyltrimethylammonium bromide (1.2 g), and deionized water (80 ml).
  4. How can the authors control the final thickness of the coated CB on the CAC surface?
  5. The authors should add more information about the used conditions and sample preparation in FE-SEM, BET, and FTIR measurements.
  6. Using a multimeter (Fluke 15b+) to measure the electrical resistance is not acceptable. The authors should use scientific methods, such as the four-probe method (Kelvin technique).
  7. The authors wrote, “Notably, the FT-IR results of CACG confirmed that CAC was evenly coated with CB [23–25]." Unfortunately, the FTIR curves in Figure 3 cannot confirm this result.
  8. In figures 3e and 3f, there aren’t control samples to compare the results before and after coating.
  9. The coating process is not clear. Can the authors provide information about the depth of the coated layer on the CAC surface? Figure 4 should be improved by adding more SEM images from different places on the coated samples.
  10. In figures 6, 7, and 8, the authors should add the standard deviations. Moreover, the authors should illustrate the number of tested samples for each experiment.
  11. The discussion and explanation for the results in Section Mechanism of the CACG Energy by Cation Effect is weak. The authors should explain in detail their results using appropriate references.
Comments on the Quality of English Language

Moderate editing of the English language is required.

Reviewer 2 Report

Comments and Suggestions for Authors

1. Please add details of dispersion of CB. Was the amount of CB and CTAB optimized?

2. What is particle size distribution of CB?

3. How do you ensure CB is still deposited on CACG after immersing in water?

4. Can the authors provide SEM images of CACG after immersing in water?

5. What is the efficiency of CACG over time?

Round 2

Reviewer 1 Report

Comments and Suggestions for Authors

The author made sufficient improvements to the manuscript.

Comments on the Quality of English Language

 Minor editing of the English language is required.